# Spectrum of Imaging Patterns of Lung Cancer following Radiation Therapy

**DOI:** 10.3390/diagnostics13203283

**Published:** 2023-10-23

**Authors:** Taylor A. Strange, Lauren T. Erasmus, Jitesh Ahuja, Rishi Agrawal, Girish S. Shroff, Mylene T. Truong, Chad D. Strange

**Affiliations:** 1Department of Pathology, University of Texas Medical Branch John Sealy School of Medicine, 301 University Blvd, Galveston, TX 77555, USA; 2Department of Anatomy and Cell Biology, Faculty of Sciences, McGill University, 845 Sherbrooke Street West, Montreal, QC H3A0G4, Canada; 3Department of Thoracic Radiology, University of Texas MD Anderson Cancer Center, 1515 Holcombe Blvd., Unit 1478, Houston, TX 77030, USA; jahuja@mdanderson.org (J.A.); ragrawal1@mdanderson.org (R.A.); gshroff@mdanderson.org (G.S.S.)

**Keywords:** lung cancer, non-small cell lung cancer, radiation therapy, radiation-induced lung injury, conventional radiotherapy, high-precision dose radiotherapy

## Abstract

Radiation therapy using conventional or newer high-precision dose techniques, including three-dimensional conformal radiotherapy, intensity-modulated radiation therapy, stereotactic body radiation therapy, four-dimensional conformational radiotherapy, and proton therapy, is an important component of treating patients with lung cancer. Knowledge of the radiation technique used and the expected temporal evolution of radiation-induced lung injury, as well as patient-specific parameters such as previous radiotherapy, concurrent chemoradiotherapy, or immunotherapy, is important in image interpretation. This review discusses factors that affect the development and severity of radiation-induced lung injury and its radiological manifestations, as well as the differences between conventional and high-precision dose radiotherapy techniques.

## 1. Introduction

Lung cancer, particularly non-small cell lung cancer (NSCLC), poses a formidable challenge to both patients and healthcare providers. Despite being the second-most-prevalent form of cancer in both men and women, it remains the deadliest. In the United States alone, there are about 200,000 new cases of lung cancer annually [1]. Late-stage diagnosis and poor prognosis are common due to the elusive nature of symptoms. Fortunately, in the United States, the number of new cases has been steadily declining, primarily due to reduced rates of smoking—the leading risk factor for lung cancer. Unfortunately, this positive trend is not mirrored in countries where smoking remains a common practice [2]. Moreover, the delayed development of lung cancer following tobacco exposure ensures that the global burden of lung cancer will persist for many years [3]. 

Treatment options for NSCLC include surgery, chemotherapy, immunotherapy, and radiation therapy (RT). While treatment regimens are tailored to individual cases, for early-stage NSCLC, surgery remains the gold standard, including newer minimally invasive techniques such as video-assisted thoracic surgery (VATS) and robot-assisted thoracic surgery (RATS) [4]. Surgical treatment can even be performed in patients with some degree of chronic pulmonary disease [5]. However, for patients who are medically inoperable (such as those with more advanced chronic pulmonary disease) or who decline surgical treatment, RT has emerged as a highly effective alternative for the treatment of early-stage NSCLC [1,6,7]. Additionally, in advanced-stage inoperable disease, RT may be administered in conjunction with chemotherapy or immunotherapy, as well as for palliation [8]. 

The landscape of RT in NSCLC has evolved significantly over the last several decades. Modern techniques of radiation therapy delivery offer enhanced precision and decreased damage to adjacent normal tissues compared with conventional radiotherapy. This review explores the gamut of newer RT modalities, including 3D conformational radiotherapy (3D-CRT), intensity-modulated radiotherapy (IMRT), stereotactic body radiotherapy (SBRT), 4D conformational radiotherapy (4D-CRT), and proton therapy. These recent advances in RT delivery offer new hope to patients for improved quality of life.

## 2. Types of Radiotherapy

### 2.1. Conventional Radiotherapy

Conventional radiotherapy, also known as 2D radiotherapy, was once the primary form of radiotherapy available for patients with NSCLC. However, it is now predominantly reserved for cases of palliation [9,10]. Conventional radiation therapy employs two parallel opposed fields that form a rectangular field of radiation [11]. Due to the large radiation field, a considerable volume of adjacent normal tissue receives radiation along with the tumor (Figure 1). Thus, when the radiation dose is being determined, the benefits of irradiating the tumor must be weighed against the risk of harm to the surrounding tissues. 

### 2.2. Three-Dimensional Conformal Radiotherapy

Three-dimensional conformal radiotherapy, or 3D-CRT, was the first of the high-precision modalities of radiotherapy to emerge. By using more beams in multiple directions, 3D-CRT allows for the radiation to conform to the shape of the tumor more closely than conventional radiotherapy [9,12]. This allows for a higher radiation dosage to the targeted tumor, as well as minimizing damage to the surrounding healthy tissue (Figure 2). Although 3D-CRT is superior to conventional radiotherapy in terms of tumor targeting and reduction in healthy tissue toxicity, further advances in RT delivery now offer even greater conformity. Thus, 3D-CRT is primarily used to treat tumors that are locally advanced and inoperable, along with postoperative radiation therapy [9]. 

### 2.3. Intensity-Modulated Radiotherapy

Intensity-modulated radiotherapy (IMRT) uses multiple beams of radiation, each with dynamic multileaf collimators, to conform more closely to the tumor compared to 2D radiotherapy or even 3D-CRT (Figure 3). This flexibility allows for better conformity to the tumor, and the dose to adjacent organs is reduced compared with 3D-CRT [9,13,14]. IMRT is commonly used in similar scenarios as 3D-CRT, but with enhanced results and less toxicity [9,12]. 

### 2.4. Stereotactic Body Radiotherapy

Stereotactic body radiotherapy (SBRT) is currently the predominant modality of high-precision radiation therapy utilized for early-stage NSCLC patients who are not surgical candidates or who refuse surgical therapy [15,16] (Figure 4). SBRT utilizes ablative techniques to target tumors, delivering a higher radiation dose per fraction over a shorter treatment duration compared with conventional RT (SBRT 1–2 weeks vs. conventional 4–6 weeks) [9,10]. SBRT must be used with caution when targeting central tumors, due to the potential for toxicity. An early trial reported toxicities and mortality for central tumors treated with SBRT, which led to a recommendation of a 2 cm “no fly” zone surrounding the central bronchial tree [17]. SBRT can be used for central lung cancers by lowering the dose per fraction. SBRT offers decreased toxicity when compared with other forms of RT. 

### 2.5. Four-Dimensional Conformal Radiotherapy

Four-dimensional conformal radiotherapy (4D-CRT) is a novel approach to delivering radiation to the chest, in that it incorporates time and patient chest movements into the treatment planning of lung tumors, which are mobile targets. Often used concurrently with 3D-CRT, IMRT, and proton therapy, 4D-CRT integrates the volumetric images of the tumor in each phase of the patient’s breathing cycle, generated by 4D-CT data acquisition. Approaches to 4D-CRT include respiratory-synchronized techniques (i.e., gating or tracking) and tumor-margin-based techniques (i.e., internal target volume or mid-position strategy). These approaches accommodate the patient’s chest movements during respiration, thereby reducing the irradiation of uninvolved tissue even more [12,13]. Further research is needed to evaluate the benefits of these techniques.

### 2.6. Proton Therapy

Proton therapy is unique compared to the other high-precision radiation techniques, in that it is particle-based rather than photon-based [9]. Consequently, it provides a very low entrance dose of radiation, which steadily increases until it reaches its peak at the targeted tumor (Bragg peak), with an abrupt drop-off in the dose distal to the tumor [9] (Figure 5). This dosage curve accounts for proton therapy’s optimal dose conformation to the tumor, with the least toxicity to the surrounding normal tissue among the high-precision radiation therapy techniques available [18,19] (Figure 6). Compared to SBRT, proton therapy is useful when targeting tumors close to critical organs such as the heart, great vessels, trachea, esophagus, and spinal cord [9,12]. Unfortunately, however, this technology is costly and only available at limited centers.

## 3. Post-Radiation Changes

Advancements in radiation therapy delivery techniques have significantly improved the prognosis and quality of life for lung cancer patients in recent years. Despite the benefits of these various forms of radiotherapy, all are associated with a spectrum of damage to healthy adjacent tissue. Post-radiation lung changes are commonly seen following both conventional and high-precision radiotherapy techniques. 

Radiation-induced lung injury (RILI) is a well-recognized consequence of radiotherapy. The extent of damage depends on various factors, including patient-specific risk factors, radiation dosage, fractionation schedules, and the type of radiotherapy received. For all types of RT, post-radiation changes are divided into two distinct stages: the acute exudative phase (also called radiation pneumonitis) in the first 6 months after the completion of RT, and the chronic fibrotic phase thereafter.

Patient-related risk factors contributing to RILI include smoking history, pre-existing pulmonary conditions such as interstitial lung disease, and previous radiotherapy to the chest. Risk is also dependent on the volume of lung tissue exposed to radiation, the radiation dose and fractionation schedule, and the concurrent use of chemotherapy [9,20,21,22,23]. This intricate interplay between clinical and treatment-related variables underscores the need for personalized approaches to synergize treatment potency, as well as to mitigate adverse effects. 

Emerging research has shed light on the impact of radiation dosage and fractionation schedules on RILI. Studies comparing the use of hyperfractionation (i.e., dispensing lower doses of radiotherapy more-than-once-daily) and hypofractionation (i.e., dispensing higher doses of radiotherapy over a shorter period of time) have highlighted the importance of optimizing the balance between tumor control and toxicity to adjacent healthy tissue [24,25]. A lower radiation dose delivered over a longer period of time reduces the biological effects of the radiation, which leads to decreased toxicity to healthy tissues but may compromise the antitumor effect. Further research is needed to establish the optimal strategy to achieve therapeutic goals while minimizing adverse events. 

Pathophysiologically, the acute exudative phase of RILI is defined by capillary injury, vascular congestion, and inflammatory cell infiltration, subsequently resulting in collagen fibril deposition [20,26] (Figure 7). Imaging manifestations of this phase predominantly include ground-glass and consolidative opacities. Other less common but well-documented imaging findings include the “crazy paving” and “reversed halo” signs. Crazy paving describes the finding of interstitial septal thickening with associated ground-glass opacities, while a reverse halo describes a crescent-shaped or peripheral rim of consolidation that surrounds a ground-glass opacity. An array of pharmaceuticals have been evaluated to mitigate lung injury related to radiotherapy. For patients experiencing symptomatic radiation lung injury, inhaled, oral, or even intravenous corticosteroids are commonly administered to provide relief from acute symptoms and potentially prevent progressive fibrotic changes [20,21,27]. 

The transition of RILI to the chronic fibrotic phase is marked by extensive fibroblastic proliferation with subsequent fibrous tissue deposition, vascular sclerosis, and ultimate collapse/obliteration of alveolar spaces, leading to tissue fibrosis in the irradiated lung [20,26]. Imaging findings during this phase reveal evolving areas of consolidation with air bronchograms and traction bronchiectasis [9,27]. 

A spectrum of imaging findings has been observed in the lungs following conventional and high-precision radiotherapy. Due to the more precise nature of radiation delivery, high-precision radiotherapy displays post-radiation changes that conform more closely to the original tumor site compared to conventional radiotherapy. Furthermore, the temporal evolution of these changes differs. For conventional radiotherapy, acute exudative changes are observed as early as 4 weeks and are usually present by 3 months. Chronic changes will peak around 6–12 months, with the changes stabilized by 12–24 months. In contrast, after high-precision radiotherapy, early findings do not typically present until 3 months post-treatment, with only about 50% of RILI patients demonstrating imaging findings at 6 months. Furthermore, many post-radiation changes after high-precision therapy continue to evolve after 2 years [1,9,23]. These long-term changes, particularly when they involve growth after 2 years, can raise concern for a recurrent tumor. 

## 4. Specific Imaging Patterns in RILI

There are a few specific imaging patterns of radiation-induced lung injury that are more commonly encountered following high-precision radiotherapy as opposed to conventional radiotherapy. These include the modified conventional pattern, the mass-like pattern, the scar-like pattern, and organizing pneumonia [9,12,13]. The first three refer to patterns described with SBRT but can also be seen with other high-precision dose techniques and occur 6 months after the RT’s completion or later. 

### 4.1. Modified Conventional Pattern

The modified conventional pattern is the most commonly encountered pattern of fibrosis identified after high-precision dose radiotherapy. It is identified in 46–71% of patients. The modified conventional pattern manifests as a focal area of consolidation, volume loss, and bronchiectasis, similar in appearance to conventional radiation changes, but is less extensive, owing to the closer conformation of radiation therapy to the tumor with the high-precision dose techniques [28]. Compared with conventional radiotherapy, air bronchograms may be less extensive and less frequent [12] (Figure 8). 

### 4.2. Mass-like Pattern

The mass-like pattern is encountered in 7–20% of patients after the completion of high-precision dose radiation therapy. On CT, this pattern presents as a focal area of fibrosis that is compact or mass-like, sometimes larger than the originally treated tumor [28] (Figure 9). As the name suggests, this pattern often closely resembles a true mass and must be distinguished from tumor recurrence. The imaging findings of the mass-like pattern are believed to be a result of the multidirectional beams for high-precision radiotherapy delivery, resulting in a more three-dimensional radiation treatment field [12,29]. Typically, this pattern lacks features like air bronchograms and straight margins [30]. Temporal evolution of shape and location (towards or away from the hilum) has been observed due to lung retraction during the first year after the completion of radiotherapy [12].

### 4.3. Scar-like Pattern

The scar-like pattern of fibrosis occurs in 11–22% of patients after the completion of high-precision dose radiation therapy. On CT, a thin linear opacity at the tumor site, usually less than 1 cm in diameter, is associated with volume loss [23,28] (Figure 10).

### 4.4. Organizing Pneumonia

Radiation-induced lung injury can present with an organizing pneumonia pattern of injury. This pattern is believed to be immunologically mediated and generally occurs between six weeks and ten months after the completion of radiotherapy [31]. Organizing pneumonia after radiation therapy has the appearance of patchy areas of ground-glass and consolidative opacities that can be migratory, with a waxing and waning course, often presenting outside of the radiation treatment area [32].

## 5. Imaging Modalities and Radiomics

Different imaging modalities are available for post-therapy follow-up. These include radiography, computed tomography, positron emission tomography/computed tomography, and magnetic resonance imaging.

### 5.1. Radiography

Chest radiography, being readily accessible and cost-effective, often serves as the initial imaging modality when patients present with respiratory symptoms following radiation therapy for lung cancer. Chest radiographs can typically detect RILI by the fourth week following the completion of radiotherapy [9]. It should be noted that radiography will usually not detect post-radiation changes unless the lung tissue received a radiation dose greater than 40 Gy [33]. During the acute phase of RILI, radiography may manifest heterogeneous opacities confined to the radiation field, while chronic fibrosis changes appear as lung consolidations confined to the radiation treatment area, with associated volume loss in the lung and traction bronchiectasis [27]. Radiography is tempered by low specificity, often necessitating cross-sectional imaging modalities for comprehensive post-radiotherapy assessment. 

### 5.2. Computed Tomography

Computed tomography (CT) is more sensitive, surpassing radiography in the early detection of post-radiation changes [1,6]. CT can detect ground-glass opacities, crazy paving, and reverse halo patterns during the acute exudative phase of radiation pneumonitis [9]. When increased soft tissue is seen within an area of radiation fibrosis, CT is limited in differentiating tumor recurrence versus infection or inflammation versus the mass-like pattern of radiation fibrosis.

### 5.3. Positron Emission Tomography/Computed Tomography

FDG (18F-2-deoxy-D-glucose) positron emission tomography/computed tomography (PET/CT) is a valuable adjunct to evaluate high-risk CT features of tumor recurrence following radiotherapy. FDG, a glucose analog, serves as a useful radiotracer due to its rapid uptake into tissues exhibiting high metabolic activity, such as malignant cells [34,35]. Therefore, PET/CT can be used to evaluate increased soft tissue within areas of radiation fibrosis to help differentiate between the mass-like pattern and tumor recurrence [7,36]. 

It is important to be aware of the limitations of PET/CT. FDG uptake is also noted in areas of infection or inflammation; therefore, increased FDG uptake is expected during the first few months following RT, due to radiation pneumonitis [1,16]. FDG-PET/CT is therefore typically deferred until at least 6 months following the completion of radiotherapy, or until after the onset of the chronic fibrotic phase of lung injury [1,16]. 

Future directions of research include the role of FDG PET/CT in the detection and quantification of radiation-induced lung injury, with volumetric PET parameters serving as potential biomarkers for assessing lung inflammation after RT. Global lung FDG uptake after RT for patients with locally advanced NSCLC can be quantified by subtracting the tumor uptake from the total lung FDG uptake using volume-based quantitative FDG PET/CT parameters such as global lung glycolysis, total lesion glycolysis, and lung parenchyma mean standardized uptake value [37].

### 5.4. Magnetic Resonance Imaging 

Magnetic resonance imaging (MRI) employs strong magnetic fields instead of X-rays to generate images, offering a promising alternative for patients who cannot tolerate iodinated contrast or who wish to minimize their radiation dose. MRI is not currently commonly used in the assessment of post-radiation changes [38]. MRI is sometimes used following lung radiotherapy in the evaluation of cardiac toxicity [39]. Recent animal studies suggest that using MRI following the administration of hyperpolarized noble gases may prove beneficial in assessing alveolar and metabolic damage in lung tissue [40]. 

### 5.5. Radiomics

Radiomics is a quantitative approach to medical imaging that uses advanced mathematical analysis to enhance current imaging data. Through this form of artificial intelligence, radiomics can quantify textural information on various imaging modalities [41]. This technique has been applied to help predict radiation-induced injury in a limited number of tumors, which may provide options for the identification and quantification of radiation-induced lung injury in the future [42].

## 6. Clinical Applications and Differential Diagnoses

Clinicians and radiologists grapple with the task of distinguishing between benign and malignant conditions, particularly in the context of imaging following radiation therapy. Knowledge of the spectrum of expected radiation-induced changes, timelines, and patterns observed with conventional and high-precision radiotherapy is important to avoid misinterpretation as infection or tumor recurrence. 

### 6.1. Distinguishing Infection from the Acute Exudative Phase of RILI

Discerning acute post-radiation changes from infection often represents a diagnostic conundrum, as both conditions may present with ground-glass opacities or consolidations. However, the temporal evolution of these changes following either conventional or high-precision radiotherapy can serve as a valuable discriminant. Ground-glass or consolidative opacities that emerge prior to the completion of radiotherapy suggest infection, rather than radiation-induced changes. The acute phase of RILI generally demonstrates gradual onset of ground-glass or consolidative opacities that are confined within the radiation treatment area, although they may breach anatomic boundaries like pleural fissures. In distinction, infectious processes, which can also present with ground-glass or consolidative opacities, develop more rapidly, are usually located outside the radiation treatment area, and typically respect anatomical borders [9,12]. 

Tree-in-bud is a pattern of centrilobular bronchial dilatation seen on thin-section chest CT, and the bronchiolar luminal filling by mucus, pus, fluid, or tumor resembles a budding tree. Tree-in-bud opacities are classically described in the setting of infectious processes (notably, typical and atypical mycobacterial disease) and aspiration [12,43]. In the post-therapy setting, tree-in-bud opacities are usually due to infection or inflammation, and less commonly due to central obstructing tumors and tumor emboli.

Cavitation can pose a diagnostic dilemma in the post-therapy setting. Cavitary lesions can be seen with typical and atypical infectious processes—primary malignancy, as well as metastasis [12,44]. It is worth noting, however, that radiation therapy, chemotherapy, targeted therapy, and immunotherapy can result in cavitation, which can be difficult to delineate from infection or tumor [23] (Figure 11). PET/CT, unfortunately, is of limited benefit, as the finding can be FDG-avid in tumors and infection/inflammation.

### 6.2. Distinguishing Tumor Recurrence from the Chronic Mass-like Pattern of RILI

The interpretation of imaging studies in patients treated with high-precision dose RT is challenging in the context of response assessment. Historically, the Response Evaluation Criteria in Solid Tumors (RECIST) were the primary framework for assessing tumor responses to therapy. First published in 2000, and subsequently revised in 2009 (now referred to as RECIST 1.1), these criteria define the evaluation of tumors following therapy, with definitions of stable disease, partial response, complete response, and progressive disease [45,46]. In RECIST 1.1., lung tumors are measured by the long-axis diameter, and progressive disease is defined as an increase in tumor size/burden of greater than or equal to 20%. On CT, the mass-like pattern of radiation fibrosis can evolve and show an increase in size beyond 2 years after RT completion. This increase in size raises suspicion for tumor recurrence and can meet the criteria for progressive disease [29,47,48]. 

To address this dilemma, Huang et al. introduced the concept of high-risk features (HRFs) delineating specific CT imaging features associated with a higher likelihood of tumor recurrence [10,15] (Table 1). Patient management is based on risk stratification: low-risk patients with no HRFs can be imaged every 3–6 months for 1 year, after which an imaging interval of 6–12 months can be considered; intermediate-risk patients with the presence of 1–2 HRFs can benefit from an FDG PET/CT (if available) and close follow-up; patients at high risk of recurrence with the presence of more than three HRFs can undergo biopsy or can proceed directly to salvage treatment (Table 2). The most sensitive and specific imaging findings suggesting tumor recurrence include enlargement more than 12 months after the completion of radiotherapy and cephalocaudal growth of more than 5 mm and/or 20% [10]. Other CT findings that are suggestive of tumor recurrence include loss of air bronchogram, the obliteration of bronchi that were previously aerated, and a bulging margin or lobulated contour of radiation changes [10,16,27,49,50] (Figure 12). 

## 7. Complications of Lung Radiotherapy

In addition to tumor recurrence, two other complications of radiation therapy for lung cancer include in situ pulmonary artery thrombosis and radiation recall pneumonitis.

### 7.1. In Situ Pulmonary Artery Thrombosis

Pulmonary emboli, commonly dislodged from the peripheral venous system, are the predominant cause of pulmonary artery obstruction. It has been noted, however, that a variety of alternative processes can promote de novo thrombosis in the pulmonary arteries, i.e., in situ pulmonary artery thrombosis [51,52]. Some of these entities include abnormal pulmonary structures (such as a stump thrombus after pneumonectomy or lobectomy), sickle-cell disease, infection, or trauma [53].

Ahuja et al. reported that in situ pulmonary artery thrombosis can also be seen as a complication of radiation therapy utilized for the treatment of lung cancer and malignant pleural mesothelioma. In a cohort of 27 patients, radiation-induced in situ pulmonary artery thrombosis was commonly solitary (93%), formed an obtuse angle with the vessel wall (89%), and was non-occlusive (96%). Additionally, all examples were eccentric and involved pulmonary arteries included in the area of radiation treatment and supplying the lung with visualized radiation changes [54] (Figure 13). It should be noted that there was no evidence of pulmonary embolus in any of the patients during follow-up, even in patients who were not treated with anticoagulation therapy. The time to the diagnosis of in situ pulmonary thrombosis from the completion of radiation therapy ranged between 53 and 2522 days, with a mean of 675 days [54].

### 7.2. Radiation Recall Pneumonitis 

Radiation recall refers to an inflammatory reaction that occurs in a previously irradiated field after exposure to a systemic agent that can occur weeks, months, or even years after the completion of radiation therapy [55]. While most cases in the literature are cutaneous, radiation recall pneumonitis has also been reported [56]. Specifically, radiation recall pneumonitis is an acute inflammatory process in the lungs (confined to the previous radiation treatment field) that occurs after the exposure to a systemic agent, such as chemotherapy, immunotherapy, targeted therapy, or even vaccinations [55,56,57] (Figure 14).

## 8. Follow-Up Recommendations

For radiologists tasked with accurately interpreting post-radiation therapy imaging, thorough knowledge of the patient’s radiation therapy type, treatment dates, and administered dosages—particularly doses exceeding 20 Gy—is paramount [9]. Radiologists should also be cognizant of the rising prevalence of high-precision radiation techniques to treat patients with lung cancer and the spectrum of findings of radiation-induced lung injury from the acute phase of pneumonitis to the chronic phase of fibrosis [58]. 

In alignment with the current recommendations, post-radiotherapy follow-up imaging should include CT scans every 3–6 months following the completion of therapy for the first year, every 6–12 months for years 1–3, and annually thereafter [10,15]. Suspicious CT findings can be evaluated with PET/CT imaging, with tissue biopsy of suspicious or persistent abnormalities [7,23,29,59]. These recommendations underscore the importance of a multidisciplinary approach to post-radiotherapy imaging, ensuring the timely and accurate identification of potential complications and facilitating patient-centric decision-making.

## 9. Conclusions

Radiation therapy is important in the treatment of patients with lung cancer. High-precision dose techniques such as three-dimensional conformal radiotherapy, intensity-modulated radiation therapy, stereotactic body radiation therapy, four-dimensional conformational radiotherapy, and proton therapy allow for the optimal radiation dose to be delivered to the tumor, improve local disease control, and reduce toxicity to adjacent tissues. Knowledge of the radiation technique used, radiation treatment plan, and expected temporal evolution of radiation-induced lung injury, along with patient-specific parameters such as previous radiation therapy, concurrent chemoradiotherapy, or immunotherapy, is important to correctly identify the expected radiological manifestations of radiation-induced lung injury and differentiate them from tumor recurrence or infection.

## Figures and Tables

**Figure 1 diagnostics-13-03283-f001:**
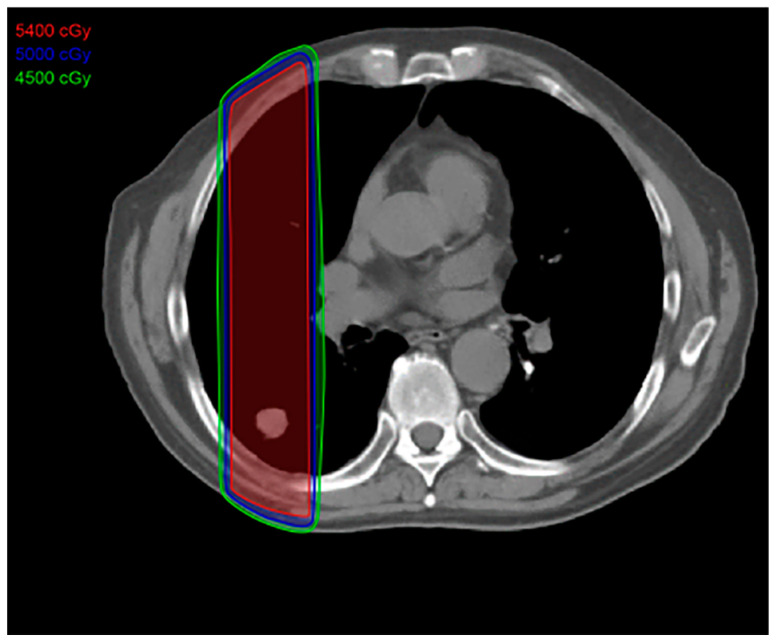
Conventional radiation treatment plan: This type of radiation therapy is easier to plan and uses an anterior posterior field that includes normal tissues in the high-dose area, resulting in increased toxicity to the adjacent normal tissues. The total dose is approximately 70 Gy, given 5 days a week for 7 weeks.

**Figure 2 diagnostics-13-03283-f002:**
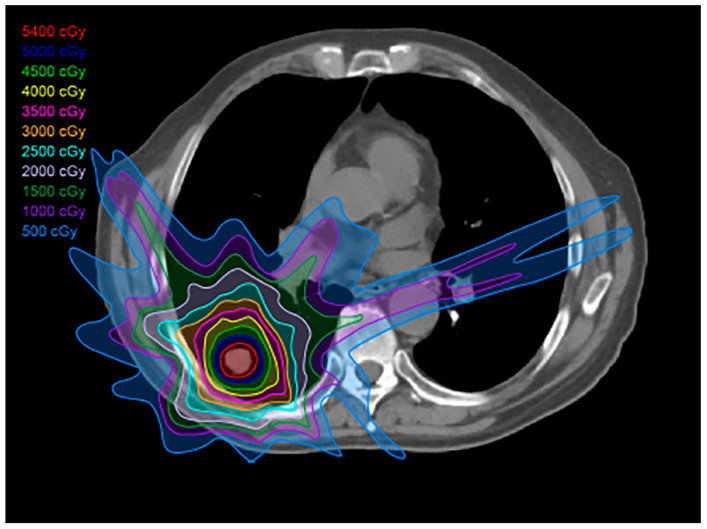
Three-dimensional (3D) conformal radiation treatment (3D-CRT) plan: This type of plan has less toxicity to the adjacent normal tissues compared to conventional radiation therapy; 3D-CRT is used to treat locally advanced, inoperable tumors, as well as in the postoperative setting.

**Figure 3 diagnostics-13-03283-f003:**
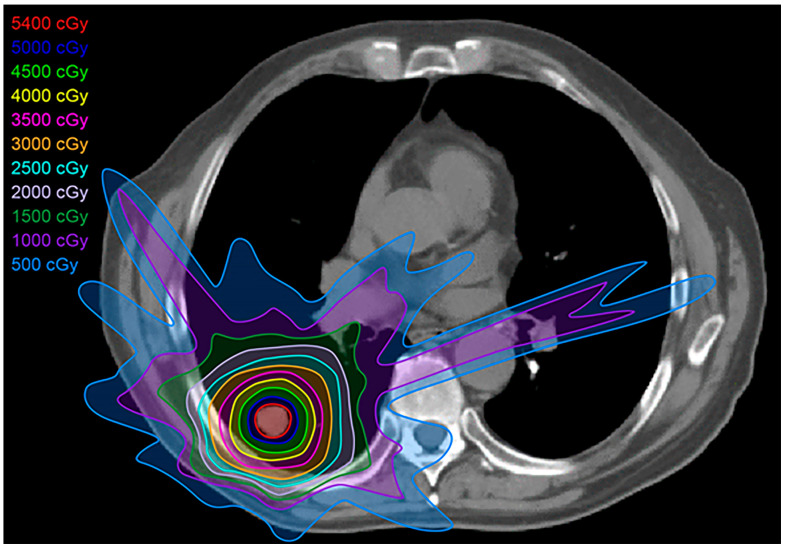
Intensity-modulated radiation treatment (IMRT) plan: IMRT is a form of 3D-CRT that uses multileaf collimators and conforms better to the shape of the tumor, with a closer margin, resulting in less toxicity to the adjacent tissues compared with conventional radiation therapy and 3D-CRT.

**Figure 4 diagnostics-13-03283-f004:**
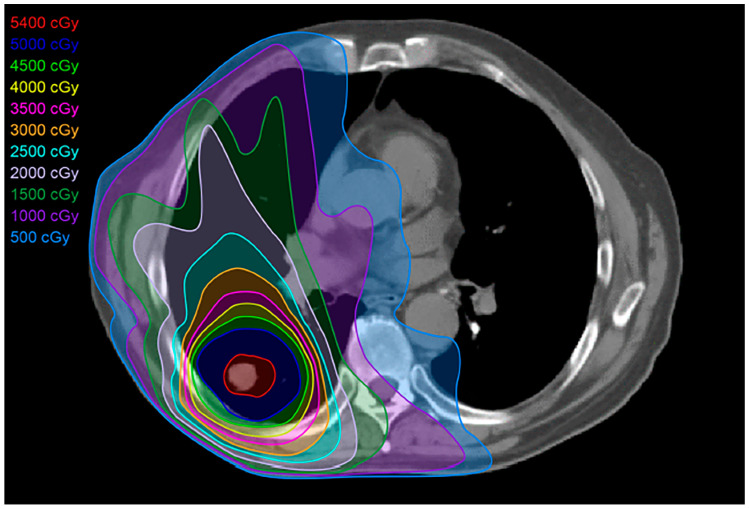
Stereotactic body radiation treatment (SBRT) plan: SBRT targets the tumor, delivering a more compact dose with a steep drop-off in all directions, thereby limiting the dose to normal adjacent tissues. SBRT’s total dose depends on the size and site of the tumor targeted. For smaller (<5 cm) peripheral tumors, the total treatment dose is approximately 25–50 Gy over 1–2 weeks. For larger, more central tumors, the total dose is 60–70 Gy.

**Figure 5 diagnostics-13-03283-f005:**
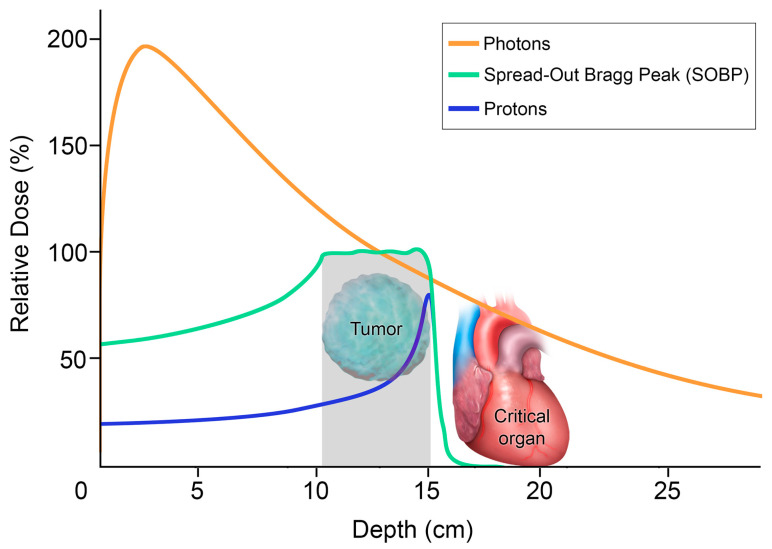
Differential depth–dose distribution of photons versus protons: The graph plots the relative dose with respect to the penetration depth. The orange is the radiation dose with photons. With no mass and no charge, photons can penetrate, but most of the radiation dose delivered is only in the first 3 cm from the patient’s skin. Photons gradually lose this energy until they reach the target and continue to deposit the dose as they exit the body, with potential collateral damage to adjacent critical organs. In contrast, protons, shown in blue for single beam and in green for multiple beams spread out, have a lower entrance dose. Protons are heavy charged particles that gradually slow down as they interact with bodily tissues. The absorbed dose increases gradually, with a sharp rise to a peak when the protons are stopped at the site of the tumor, where there is a large deposit of energy. This is known as the Bragg peak. The abrupt decrease in dose distal to the tumor target yields a minimal exit dose, sparing the adjacent critical organs.

**Figure 6 diagnostics-13-03283-f006:**
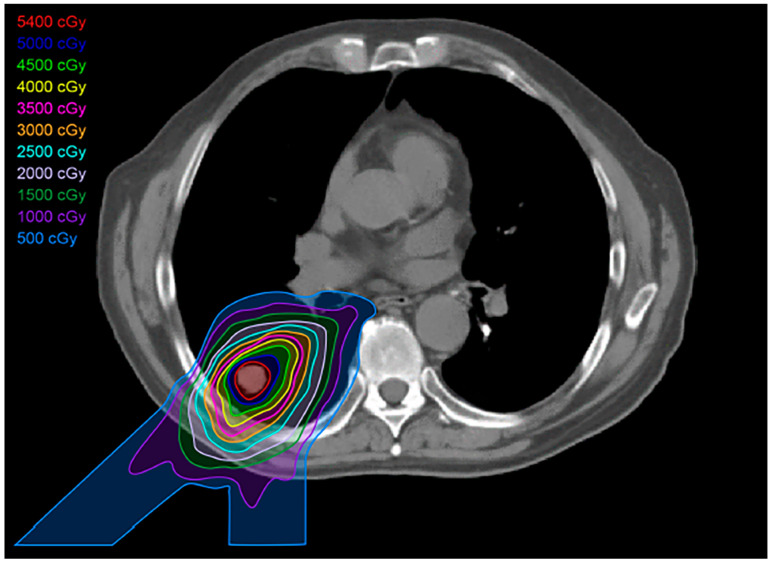
Proton treatment plan: Proton beams show a lower dose at skin entry, a sharp rise to a Bragg peak where the maximum dose is deposited directly into the tumor, and an almost non-existent exit dose. The reduced volume of healthy tissue exposed to intermediate and low doses of proton therapy results in less dose to organs at risk.

**Figure 7 diagnostics-13-03283-f007:**
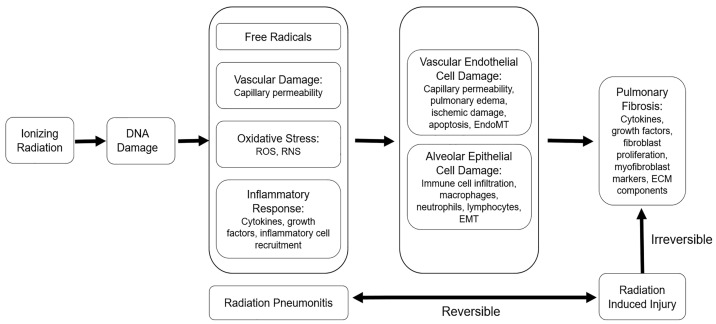
The pathophysiology of radiation injury: Ionizing radiation induces free radicals and DNA damage to promote oxidative stress, vascular damage, and inflammation, which manifest as radiation pneumonitis. Persistent inflammation sustains alveolar epithelial and vascular endothelial cell damage and contributes to pathological changes, including immune cell infiltration, capillary permeability, and pulmonary edema. Prolonged alveolar and vascular damage lead to epithelial-to-mesenchymal transition (EMT) and/or endothelial-to-mesenchymal transition (EndoMT) and eventually culminate in fibrotic changes. Adapted from Hanania, AN. Chest 2019, (156): 150 [1].

**Figure 8 diagnostics-13-03283-f008:**
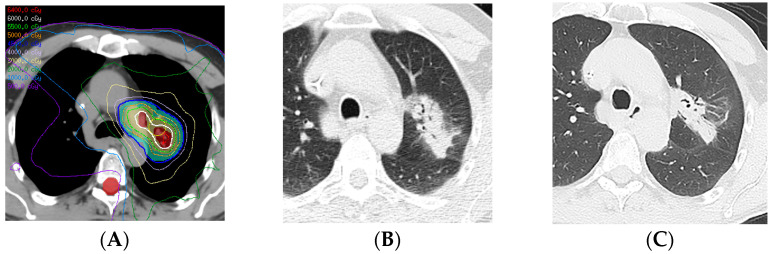
Modified conventional pattern of lung injury following IMRT (60 Gy, 15 fractions): (**A**) Computed dosimetric axial reconstruction obtained for an IMRT plan, showing the central left upper lobe tumor and left mediastinal lymph node receiving the maximal isodose (64 Gy). (**B**) CT image 6 months after completion of RT, showing consolidative opacity with air bronchograms, consistent with radiation pneumonitis. (**C**) CT image 7 years after RT, showing consolidative opacity, volume loss, and traction bronchiectasis, similar to but less extensive than fibrosis seen with conventional radiation therapy.

**Figure 9 diagnostics-13-03283-f009:**
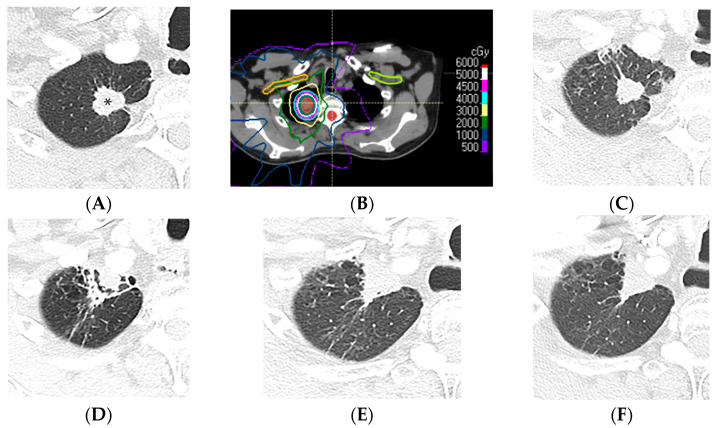
Mass-like pattern of lung injury after SBRT (70 Gy, 10 fractions): (**A**) CT before treatment, showing a right upper lung cancer (*). (**B**) Computed dosimetric axial reconstruction obtained for the SBRT plan, showing the primary lung malignancy receiving the maximal isodose (77 Gy). (**C**) CT image 3 months after completion of RT, showing that the malignancy has slightly decreased in size, with minimal radiation pneumonitis in the adjacent lung. (**D**) CT at 9 months after RT, showing retraction of the treated primary tumor anteriorly abutting the anterior pleura, surrounded by a peripheral solid rim with normal interspersed lung tissue forming a target sign. (**E**,**F**) CT images 3 and 4 years after RT, respectively, showing a triangular mass-like opacity. Note that evolution of the radiological findings >2 years after SBRT completion is common, and mass-like fibrotic changes can occur up to 4 years after and be misinterpreted as recurrence of malignancy.

**Figure 10 diagnostics-13-03283-f010:**
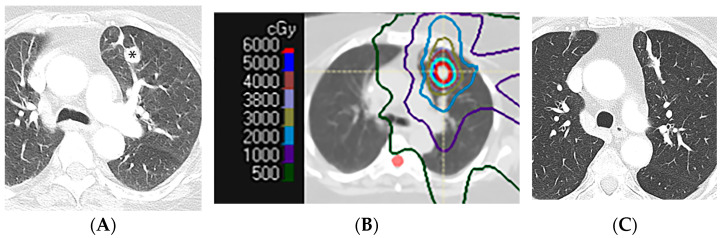
Scar-like pattern of lung injury after SBRT (54 Gy, 3 fractions): (**A**) CT before treatment, showing a left upper nodule (*). (**B**) Computed dosimetric axial reconstruction obtained for the SBRT plan, showing the primary lung malignancy receiving the maximal isodose (65 Gy). (**C**) CT image 36 months after completion of RT, showing a linear opacity in the radiation treatment plan, consistent with a scar-like pattern of radiation injury.

**Figure 11 diagnostics-13-03283-f011:**
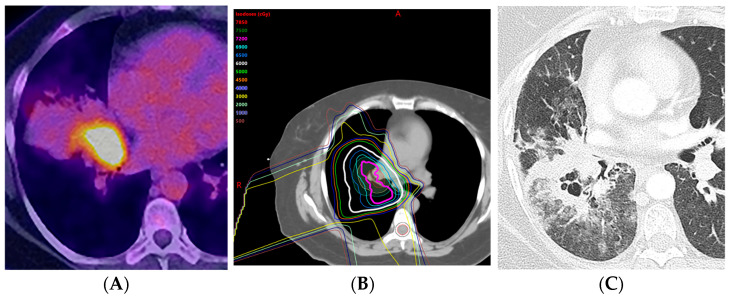
Acute radiation pneumonitis in a patient treated with concurrent chemotherapy (proton therapy: 72 Gy, 30 fractions) followed by immunotherapy with durvalumab: (**A**) PET/CT showing the FDG-avid primary tumor in the central aspect of the right middle lobe, associated with post-obstructive pneumonia. (**B**) Computed dosimetric axial reconstruction obtained for the proton plan, showing the primary malignancy receiving the maximal isodose. (**C**) CT 3 months after proton therapy completion, showing ground-glass and cavitary consolidative opacities adjacent to the treated primary tumor. A spectrum of radiation injury findings in the acute phase is seen with both conventional RT and high-precision dose techniques such as proton therapy. When the consolidative and GG opacities are greater than 5 cm, this is considered diffuse; when they are less than or equal to 5 cm, this is considered patchy.

**Figure 12 diagnostics-13-03283-f012:**
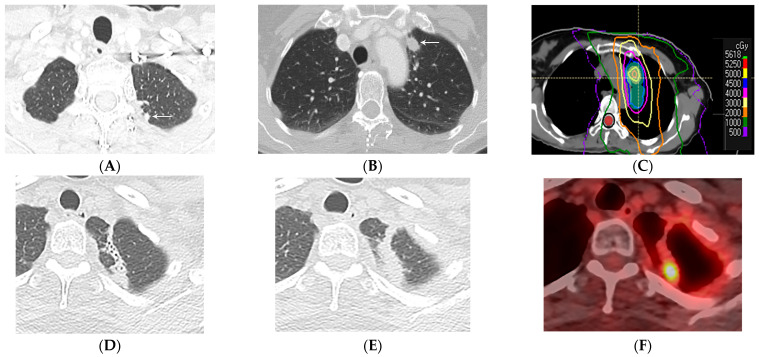
Local recurrence of malignancy after IMRT (52.5 Gy, 15 fractions) to the left upper lobe: (**A**,**B**) CT before treatment showing a 1.5 cm left upper nodule (arrow) in (**A**) and a biopsy-proven satellite 0.6 cm nodule in the left apex posteriorly (arrow) in (**B**); metastatic disease to the bone is not shown. (**C**) Computed dosimetric axial reconstruction obtained for IMRT planning, showing a malignancy in the left apex receiving the maximal isodose (56 Gy). (**D**) CT 1 year after RT showing a focal consolidative opacity with traction bronchiectasis, associated with volume loss and architectural distortion in a scar-like pattern of fibrosis. (**E**) CT 2 years after RT showing loss of visualization of the air bronchogram in the area of fibrosis suspicious for tumor recurrence. (**F**) PET/CT showing FDG-avid focus within the enlarging opacity at the treated site, suspicious for local tumor recurrence. Biopsy showed adenocarcinoma. Note that the loss of visualization of air bronchograms, enlarging opacity at the treated site, and enlargement after 12 months are high-risk features on CT, predictive of local recurrence.

**Figure 13 diagnostics-13-03283-f013:**
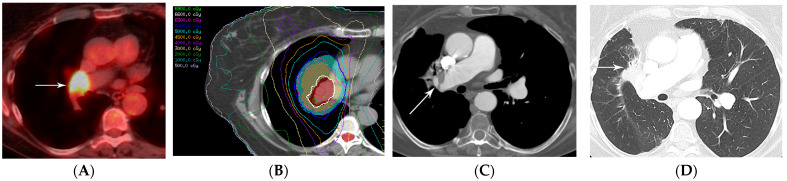
In situ thrombus of the pulmonary artery in a patient treated with IMRT (66 Gy, 30 fractions): (**A**) PET/CT showing the FDG-avid primary tumor in the central aspect of the right middle lobe (arrow). (**B**) Computed dosimetric axial reconstruction obtained for the SBRT plan, showing the right middle lobe central tumor receiving the maximal isodose (69 Gy). (**C**) CT with intravenous contrast 5 years after RT completion, showing an eccentric, non-occlusive thrombus in the right pulmonary artery (arrow). (**D**) CT showing radiation fibrosis in the right perihilar region (arrow). Typical of in situ thrombosis, the pulmonary artery was within the radiation treatment plan and the supplied lung showed radiation fibrosis.

**Figure 14 diagnostics-13-03283-f014:**
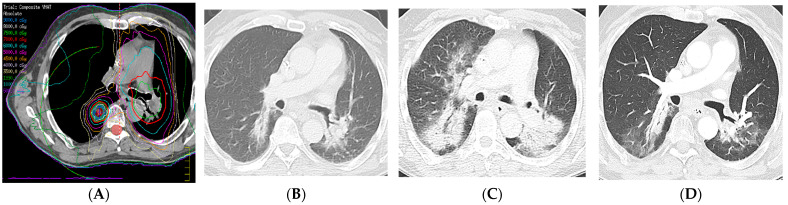
Radiation recall pneumonitis (RRP): Four years earlier, the patient presented with left upper lobe squamous-cell cancer with nodal metastases to the contralateral mediastinum and was treated with intensity-modulated radiation therapy (IMRT) (66 Gy, 30 fractions) to the left upper lobe and mediastinum. Two years earlier, the patient presented with a right lower lobe squamous-cell cancer and was treated with IMRT (70 Gy, 10 fractions). (**A**) Composite computed dosimetric axial reconstruction showing both IMRT plans. (**B**) CT image 1 year after completion of the second round of radiation therapy and before the start of immunotherapy with pembrolizumab, showing bilateral lower lobe radiation fibrosis posteriorly. (**C**) CT image when the patient presented with shortness of breath and cough 5 months after the start of immunotherapy, showing new bilateral airspace opacities that conform to the radiation treatment plan, consistent with radiation recall pneumonitis; symptoms improved after steroid therapy and temporarily withholding immunotherapy. (**D**) CT 2 weeks following steroid tapering, showing pulmonary findings of radiation fibrosis similar to that from before the start of immunotherapy. RRP refers to inflammation in previously irradiated lung tissue after exposure to an inciting pharmacological agent (e.g., immunotherapy).

**Table 1 diagnostics-13-03283-t001:** CT findings predictive of local recurrence of malignancy in the radiated lung after SBRT (adapted from Hanania, AN. Chest 2019, (156): 150) [1].

Enlarging Opacity at the Primary Site
Sequential enlargement
Enlargement 12 months after RT completion
Bulging margins
Linear margin disappearance
Loss of air bronchogram
Craniocaudal growth of ≥5 mm and ≥20%

**Table 2 diagnostics-13-03283-t002:** Practical management guidelines to evaluate for local recurrence of malignancy after SBRT.

Risk Level	Number of High-Risk Features	Management
Low	0	CT every 3 to 6 months for 1 year, and every 6 to 12 months thereafter
Intermediate	1–2	CT at 3 months or FDG PET/CT
High	≥3	Biopsy or salvage therapy

## Data Availability

Not applicable.

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
