# Peer review of "Spectrum of Imaging Patterns of Lung Cancer following Radiation Therapy"

_diagnostics, 2023, doi:10.3390/diagnostics13203283_

Round 1
Reviewer 1 Report
This is a comprehensive review of the types of radiotherapy for lung cancer and the changes that occur after radiotherapy.
Although clinically very important and necessary to determine the effectiveness of treatment and to detect serious adverse events, there has been limited organized literature on the subject. Therefore, I think this paper is very useful in clinical practice.
In particular, the fact that post-treatment changes are different in high-precision therapy compared to conventional therapy is a finding that should be widely known (and sometimes explained to patients) by medical professionals (not only radiotherapists but also oncologists and respiratory physicians) involved in current lung cancer treatment.
Please confirm the following six points.
1) Page 3, Figure 2
Figure 2 seems to be incorrect. (Isn't what the author has posted an inverted IMRT figure?)
2) Page 3, line 83.
It may be more appropriate to mention that the dose to adjacent organs is reduced in intensity-modulated radiotherapy, but the low-dose area in the lung field may be expanded.
3) Page 4, line 94.
Please consider adding that stereotactic radiotherapy is also used for central lung cancer by lowering the dose per fraction.
4) Page 5, line 123.
Please add carbon-ion radiotherapy as a treatment modality. Or, change the proton therapy section to particle therapy and list both proton and carbon-ion radiotherapy. Please refer to the following paper and others.
Kubo N, et al. Clinical results of carbon ion radiotherapy for inoperable stage I non-small cell lung cancer: A Japanese national registry study (J-CROS-LUNG). Radiother Oncol. 2023 Jun:183:109640. doi: 10.1016/j.radonc.2023.109640. Epub 2023 Mar 27.
Okano N, et al. Clinical results of carbon-ion radiotherapy for stage I non-small cell lung cancer with concomitant interstitial lung disease: a Japanese national registry study (J-CROS-LUNG). Journal of Radiation Research, Volume 64, Issue Supplement_1, June 2023, Pages i2–i7, https://doi.org/10.1093/jrr/rrad008
Saitoh JI, et al. A phase I study of Hypofractionated Carbon-ion Radiotherapy for Stage III Non-small Cell Lung Cancer. Anticancer Res. 2018;38(2):885–91. doi: 10.21873/anticanres.12298.
5) Page 17, line 501.
Is the use of contrast material recommended for follow-up CT?
6)
There are two periods(..) at the end of the legends in Figures 8 and 11.
Author Response
1) Page 3, Figure 2
Figure 2 seems to be incorrect. (Isn't what the author has posted an inverted IMRT figure?)
Reply: The figure was inverted. We have fixed the figure.
2) Page 3, line 83.
It may be more appropriate to mention that the dose to adjacent organs is reduced in intensity-modulated radiotherapy, but the low-dose area in the lung field may be expanded.
Reply: We have revised the manuscript to add this point.
"This flexibility allows better conformity to the tumor and the dose to adjacent organs is reduced compared with 3D-CRT."
3) Page 4, line 94.
Please consider adding that stereotactic radiotherapy is also used for central lung cancer by lowering the dose per fraction.
Reply: We have revised the manuscript to add this statement.
"SBRT can be used for central lung cancers by lowering the dose per fraction."
4) Page 5, line 123.
Please add carbon-ion radiotherapy as a treatment modality. Or, change the proton therapy section to particle therapy and list both proton and carbon-ion radiotherapy. Please refer to the following paper and others.
Kubo N, et al. Clinical results of carbon ion radiotherapy for inoperable stage I non-small cell lung cancer: A Japanese national registry study (J-CROS-LUNG). Radiother Oncol. 2023 Jun:183:109640. doi: 10.1016/j.radonc.2023.109640. Epub 2023 Mar 27.
Okano N, et al. Clinical results of carbon-ion radiotherapy for stage I non-small cell lung cancer with concomitant interstitial lung disease: a Japanese national registry study (J-CROS-LUNG). Journal of Radiation Research, Volume 64, Issue Supplement_1, June 2023, Pages i2–i7, https://doi.org/10.1093/jrr/rrad008
Saitoh JI, et al. A phase I study of Hypofractionated Carbon-ion Radiotherapy for Stage III Non-small Cell Lung Cancer. Anticancer Res. 2018;38(2):885–91. doi: 10.21873/anticanres.12298.
Reply: We thank the reviewer for bringing this to our attention. Carbon ion is an exciting technology available in Asia and Europe. As we have no experience with carbon ion, we do not want to mislead the reader that these manifestations of radiation injury can be seen with carbon ion. Our review is based on the cases that we see in clinical practice and at this time, we respectfully decline to include carbon ion.
5) Page 17, line 501.
Is the use of contrast material recommended for follow-up CT?
Reply: The use of contrast material is at the discretion of the institution. Our hospital uses intravenous contrast on follow-up CT. However, at other academic institutions in the United States, the follow-up CT is performed without intravenous contrast.
6)
There are two periods(..) at the end of the legends in Figures 8 and 11.
Reply: There is one period at the end of the legends in Figures 8 and 11.
Reviewer 2 Report
This is a good review for learning, which briefly describes the radiation-induced lung injury pathology and its radiologic features. Despite this topic is published and updated every year, I consider this one as an excellent tool for non-radiobiologists and non-radiologists to understand this complex undesired side effect of radiation therapy for lung cancer. It is well written and very easy to read.
Author Response
We thank the reviewer for these comments. No revision was requested.